# Identification and Specific KASP Marker Development for Durum Wheat T2DS-2AS.2AL Translocation Line YL-429 with Wax Inhibitor Gene *IW2*

**DOI:** 10.3390/plants14071077

**Published:** 2025-04-01

**Authors:** Sujie Yang, Fan Yang, Zujun Yang, Wenjing Hu, Hongxia Ding, Feiyang Yang, Hongshen Wan, Zehou Liu, Tao Lang, Ning Yang, Jie Zhang, Yun Jiang, Junyan Feng, Hao Tang, Qian Chen, Qian Deng, Ying Wang, Jingyu Wu, Jun Xiao, Xing Fan, Yonghong Zhou, Jun Li

**Affiliations:** 1Biotechnology and Nuclear Technology Research Institute, Sichuan Academy of Agricultural Sciences, Chengdu 610066, China; sue123chocolate@msn.com (S.Y.); yfwheat@scsaas.cn (F.Y.); dinggzy@scsaas.cn (H.D.); yangfeiyang@scsaas.cn (F.Y.); langtao@scsaas.cn (T.L.); zhang_jie@scsaas.cn (J.Z.); jiangyun@scsaas.cn (Y.J.); fengjunyan@scsaas.cn (J.F.); chenqian8306@scsaas.cn (Q.C.); dengqian@scsaas.cn (Q.D.); wangyinggzy@scsaas.cn (Y.W.); wujingyu@scsaas.cn (J.W.); xiaojun1996@scsaas.cn (J.X.); 2College of Agronomy, Sichuan Agricultural University, Chengdu 611130, China; 3Crop Research Institute, Sichuan Academy of Agricultural Sciences, Chengdu 610066, China; wanhongshen@scsaas.cn (H.W.); liuzh@scsaas.cn (Z.L.); yangning@gmail.com (N.Y.); tanghaosc@scsaas.cn (H.T.); 4Triticeae Research Institute, Sichuan Agricultural University, Chengdu 611130, China; fanxing9988@163.com; 5School of Life Science and Technology, University of Electronic Science and Technology, Chengdu 611731, China; yangzujun@uestc.edu.cn; 6Lixiahe Institute of Agricultural Sciences, Yangzhou 225007, China; huren2008@126.com

**Keywords:** alien introgression, karyotype analysis, chromosome translocation, genome resequencing, glaucousness

## Abstract

Non-glaucous wheat can reduce solar light reflection in low-light cultivation regions, enhancing photosynthetic efficiency and potentially increasing yield. In previous work, a non-glaucous cuticular line, YL-429, was discovered in derivatives of pentaploid hybrids by crossing the synthetic wheat LM/AT23 (non-glaucous cuticular) with its tetraploid donor parent LM (glaucous) and selfing to F_7_ generations. In the present study, multicolor fluorescence in situ hybridization was used to characterize the karyotype of the YL-429 line; genome resequencing was performed to identify the breakpoint of the 2D-2A chromosome translocation of YL-429; and bulk sequencing analysis was conducted to detect the SNP in the translocated fragment and accordingly develop specific kompetitive allele-specific PCR markers for use in breeding. The line YL-429 was preliminarily determined as a 2DS and 2AS translocation (LM T2DS-2AS.2AL) line through karyotyping. Genome alignment identified an approximately 13.8 Mb segment, including the wax inhibitor gene *Iw2*, in the telomeric region of the 2DS chromosome arm replacing an approximately 16.1 Mb segment in that of the 2AS chromosome arm. According to the bulk DNA sequencing data, 27 specific KASP markers were developed for detecting the translocated fragment from the 2DS of *Aegilops tauschii*. The LM T2DS-2AS.2AL translocation line YL-429 could be helpful in improving the photosynthesis of durum wheat cultivated in low-light cultivation regions. The developed markers can assist the screening of the T2DS-2AS.2AL translocation in breeding.

## 1. Introduction

Durum wheat (*Triticum turgidum* L. ssp. *durum*, 2*n* = 4*x* = 28, AABB) plays a crucial role in national food security and human nutrition [1]. Annually, durum wheat accounts for 8–10% of the global wheat cultivation area [2,3]. Durum wheat is commonly used to produce high-protein, elastic, and cooking-resistant pasta [4]. Compared to bread, cooked pasta provides a medium-to-low glycemic index, helping consumers maintain stable blood sugar levels after consumption [5,6,7,8,9]. As a result, with improving living standards and the promotion of healthy eating concepts, the market demand for durum wheat products rich in protein and dietary fiber has been steadily increasing. Durum wheat breeding is mainly conducted through traditional methods such as pedigree selection, backcrossing, pure line selection, and recurrent selection [10,11,12]. Durum wheat (AABB genome) can incorporate superior traits from common wheat (AABBDD genome) through hybridization. In recent years, with advancements in genomics and molecular breeding technologies, molecular markers have been found to significantly reduce the time required for breeding [13,14]. Studies have demonstrated that molecular-marker-assisted selection has achieved success in improving disease resistance and quality traits. For instance, marker-assisted backcrossing using simple sequence repeats markers has been employed to enhance the grain protein content in wheat [15]. Additionally, the use of kompetitive allele specific PCR (KASP) markers in marker-assisted selection has improved durum wheat yield in heat-stressed regions and enhanced adaptability to climate change [16].

Glaucous cuticular is a collective term for the lipid components on the surface of the plant cuticle, forming crystal-like coverings on the plant’s cuticle surface resembling a white frost [17,18]. It plays a crucial role in various physiological functions and developmental processes in plants, such as limiting transpiration, reducing non-stomatal water loss, protecting plants from UV damage, and resisting pests and diseases [18,19,20,21,22,23,24,25,26]. However, non-glaucous cuticular durum wheat can reduce solar reflection in low-light regions, enhancing photosynthesis, which is beneficial for the synthesis of photosynthetic products during the grain-filling period and significantly increases yield [27]. The glaucous cuticular phenotype of the wheat epidermis is mainly controlled by two sets of dominant genes, including the glaucous cuticular formation gene *Wax1* (*W1*) and glaucous cuticular inhibitor gene (*Iw1*) located on chromosome 2BS, as well as *W2* and *Iw2* on chromosome 2DS [28,29,30,31,32,33,34,35]. The presence of a single *Iw* allele in the genome can suppress the function of *W1* or *W2* genes, resulting in a non-glaucous cuticular phenotype [26,29]. The *Iw3* and *Ws* loci, which regulate the glaucous cuticular phenotype on the spike, have been mapped to 1BS and 1AS, respectively [36].

Previously, a non-glaucous cuticular synthetic wheat LM/AT23 (AABBDD) was developed by crossing a glaucous cuticular durum wheat (AABB) with a non-glaucous cuticular *Aegilops tauschii* Coss. ssp. *strangulata* accession [37]. The glaucous cuticular character of durum wheat should be inhibited by the inhibitor *IW2* in *Ae. tauschii* [34,37]. Then, LM/AT23 was hybridized with its tetraploid donor parent (glaucous cuticular) to produce F_1_ pentaploid hybrids and self-pollinated to the high generation. A mutant line, YL-429, was identified, which exhibited a non-glaucous cuticular phenotype in the stem, leaves, and spike epidermis. In this study, the chromosomal composition and translocation breakpoint identification of YL-429 was performed through multicolor fluorescence in situ hybridization (MC-FISH) and genomic resequencing, and specific KASP markers were developed for subsequent breeding applications.

## 2. Results

### 2.1. Karyotype Characterization

In the first round of hybridization, strong signals from the Oligo-pTa535-1 probe were detected in the subtelomeric and centromeric regions of the 2AS chromosome arm of LM and YL-429 (Figure 1a,c). Moreover, a green signal from the Oligo-pSc119.2-1 probe was observed in the telomeric region of the 2AS chromosome arm of the YL-429 line (Figure 1a,c). The 2DS chromosome arm of *Ae. tauschii* AT23 showed a similar probe hybridization signal to the YL-429 line in the telomeric and subtelomeric regions, and a red signal from the Oligo-pTa535-1 probe was observed in 2DL, except at the centromere (Figure 1a,c). In the second round of hybridization, the AL-1 signal was observed in the telomeric region of the 2AS chromosome arm of LM but not in that of the YL-429 line (Figure 1a,c).

According to the hybridization position of the probes (Figure 1b) [38], Oligo-pTa535-1, Oligo-pSc119.2-1, and AL-1, in the physical maps (reference genome: CS V.1.0), the YL-429 line was speculated to be an LM T2DS-2AS.2AL translocation line, in which a small segment (about 17 Mb) of 2DS from *Ae. tauschii* AT23 replaced the entire arm or a small segment of 2AS chromosome arm of durum wheat LM.

### 2.2. Breakpoint Identification Through Resequencing

After quality filtering, a total of 570.51Gb bases were generated. By aligning the resequencing data with the merged reference genome, the sequencing depth of each chromosome in LM T2DS-2AS.2AL translocation line YL-429 was calculated (Figure 2a). A mean 35× sequencing depth was detected in both the A and B subgenomes of YL-429, except for the telomeric region of the 2AS chromosome arm; only the telomeric region of the 2DS chromosome arm exhibited high depth of reads coverage in the D genome (Figure 2b). Single-chromosome alignment revealed a 16.1 Mb segment deletion (containing 894 genes, including 327 high-confidence genes and 567 low-confidence genes) in the telomeric region of the 2AS chromosome and a 13.8 Mb segment addition (containing 613 genes, including 350 high-confidence genes and 263 low-confidence genes) from the telomeric region of the 2DS chromosome in YL-429 (Figure 2c). Therefore, the YL-429 line was identified as an LM T2DS-2AS.2AL small segment translocation line.

### 2.3. Identification of Specific KASP Markers

Glaucous and non-glaucous individuals of YL-724 × Chuanmai98 F_2_ populations were selected to construct the phenotypically contrasting bulks. YL-724 and Chuanmai98 were also used to construct the parental bulks. After quality control, the clean reads of each sample achieved a Q30 value of over 92%, and the mapping rate against the reference genome Chuanmai104 exceeded 96%, indicating that the sequencing data obtained from the bulked segregation analysis (BSA) were of high quality and suitable for subsequent analysis. The candidate wax inhibitor gene was mapped on the 0.9–17.2 Mb region of the 2DS chromosome, which covered the translocated fragment from *Ae. tauschii* in the mutant line YL-429. Through alignment with the reference genome Chuanmai104, 131 candidate SNPs were identified in the 0–13.8 Mb translocated region of the 2DS chromosome arm. Among these, 34 evenly distributed SNPs were selected to design KASP markers. Twenty-seven polymorphic markers (Appendix A) formed clear and specific genotyping clusters, while the remaining markers failed to form distinct clusters. In the genotyping results for these 27 KASP markers, LM, containing only the AB genome, showed no amplification of specific alleles, consistent with the negative control and confirming its lack of chromosome 2D. In contrast, YL-429 and YL-724 were detected as the same genotype at all 27 markers, while Chuanmai98 was the other genotype. These results validate the linkage specificity of KASP markers for the translocated segment of the 2DS chromosome arm.

## 3. Discussion

Cytogenetic analysis and genome resequencing speed up the identification of chromosome composition in variant materials. In the present study, the non-glaucous cuticular line YL-429 was visually determined to be an LM T2DS-2AS.2AL chromosome translocation line using fluorescence in situ hybridization, an approach that plays a critical role in characterizing the karyotype composition and understanding the visible variations in phenotypes or other traits [39,40,41]. The breakpoint of translocated chromosomes can be preliminarily identified by mapping the physical position of hybridization probes [38]. In this study, the 2D chromosome breakpoint of the LM T2DS-2AS.2AL line was quickly detected using three probes, involving an approximately 17 Mb segment addition from *Ae. tauschii* AT23. The line was further accurately identified using genome resequencing, revealing the addition of a 13.8 Mb segment from *Ae. tauschii* and the deletion of a 16.1 Mb segment from durum wheat LM. This demonstrates that cytological analysis is of great significance for the convenient, rapid, and low-cost identification of variants with large-scale chromosomal mutations in wheat, especially for primary breeding materials or populations. Advanced sequencing technologies further make up for the deficiency of cytological analysis in identifying mutants at the molecular level and provide more precise identification results for further research.

The LM T2DS-2AS.2AL translocation line YL-429 impairs epicuticular wax deposition by inhibiting the synthesis of wax intermediates. The 13.8 Mb translocation segment in the developed line was found to contained *Iw2*, a wax inhibitor gene mapped on chromosome 2DS, between 7.5 Mb and 8.3 Mb of the reference genome CS [29,42] leading to the failure to form a glaucous coating on any of the aerial organs. The biosynthesis of epidermal wax primarily involves the de novo biosynthesis of C16 or C18 fatty acids in plastids and the elongation of these fatty acids to produce saturated very-long-chain fatty acids in the endoplasmic reticulum. Finally, various derivatives are synthesized and embedded into the wax, including alkanes, aldehydes, primary alcohols, secondary alcohols, unsaturated fatty acids, ketones, and wax esters, as well as triterpenes, sterols, and flavonoids [43,44,45,46,47,48]. The genes associated with the wax biosynthetic process may be down-regulated in the LM T2DS-2AS.2AL line YL-429.

The non-wax LM T2DS-2AS.2AL line YL-429 has great potential in wheat breeding. Epicuticular wax on the surface of aerial plants prevents non-stomatal water loss, protects organs from UV radiation damage, and imposes a physical barrier against pathogenic infection water loss [18,19,34,45,47]. However, in environments with short daylengths or insufficient light intensity, such as low-light cultivation conditions, wheat varieties lacking cuticular wax may enhance their yield by reducing solar reflection and improving photosynthetic efficiency [49]. Therefore, the LM T2DS-2AS.2AL line YL-429 identified in this study could be used for the genetic improvement of durum wheat varieties in regions with limited sunlight. In addition, the impaired epicuticular wax deposition can significantly increase cuticle permeability due to water loss and chlorophyll efflux [50]. These developed KASP markers for the LM T2DS-2AS.2AL line can be used for tracking translocated segments in breeding programs, thereby enhancing breeding efficiency.

In conclusion, by integrating cytogenetic and whole-genome resequencing techniques, the present research involved the systematic characterization of a novel non-glaucous durum wheat genetic material, LM T2DS-2AS.2AL translocation line YL-429, which was introgressed with an alien D genome fragment containing the wax inhibitor gene *Iw2*. Moreover, specific KASP markers were developed for further molecular marker-assisted breeding. For the LM T2DS-2AS.2AL translocation line YL-429, we are constructing population of recombinant inbreed lines for the fine mapping of the wax inhibitor gene in combination with gene expression and genome resequencing results.

## 4. Materials and Methods

### 4.1. Plant Materials

YL-429, a non-glaucous epidermis line, was produced by crossing the synthetic hexaploid wheat YL-724 with its AB genome parent LM and selfing the pentaploid hybrids to the F_7_ generation (Figure 3) [37]. The amphiploid line YL-724 (2*n* = 6*x* = 42, AABBDD) with a non-glaucous epidermis was synthesized by Dr. Wuyun Yang by crossing durum wheat LM (*Triticum turgidum* ssp. *durum*, 2*n* = 4*x* = 28, AABB) with the diploid *Ae. tauschii* accession AT23 (*Aegilops tauschii* Coss. ssp. *strangulata*, 2*n* = 14, DD); the line was maintained at the Sichuan Academy of Agricultural Sciences (SAAS), China.

Chuanmai98 (2n = 6x = 42, AABBDD) is a synthetic wheat-derived hexaploid cultivar with glaucous epidermis provided by Dr. Wuyun Yang. By crossing YL-724 with Chuanmai98, an F_2_ phenotypic segregation population of epicuticular wax was generated for BSA sequencing and KASP marker development.

### 4.2. Cytogenetic Analysis

Fluorescence in situ hybridization is an important technology used in studying variations in chromosome number and structure. MC-FISH was used to investigate the chromosomal structural variations in YL-429. The seeds of LM, AT23, LM/AT23 amphiploid line YL-724, and YL-429 were germinated at 25 °C for 2–3 days. The chromosome spreads were prepared using the methods as described by Han et al. [51]. The MC-FISH procedure was performed as described by Tang et al. [52] and Fu et al. [53]. The oligonucleotide probes Oligo-pSc119.2-1 and Oligo-pTa535-1, developed by Tang et al. [52], were used in the first round of MC-FISH.

To preliminarily determine the breakpoint of the 2D-2A chromosome translocation of YL-429, the second round of hybridization was performed on the same microscope slide. After rinsing the slide with 2 × saline sodium citrate buffer, the previous hybridization probes were removed, and the probe AL-1 designed by Pro. Zujun Yang, which specifically hybridizes with the telomeric region (0–200 Kb) of the 2AS chromosome arm, was hybridized with the same metaphase cell (synthesized by Shanghai Invitrogen Biotechnology Co. Ltd., Shanghai, China). The probe sequence was GGACG TTTAG TAATT CTAAA AGGGA CTACA GGATG AAAAA CAAAA.

### 4.3. Genome Resequencing

The genome resequencing was carried out on YL-429 to accurately identify the chromosomal breakpoint. Leaf samples were collected at the four-leaf stage. Genomic DNA was extracted using the NuClean Plant Genomic DNA Kit (CWBio, Beijing, China) and sent to the DNA Stories Bioinformatics Center, Chengdu, China for sequencing. After a DNA quality assessment, the samples were randomly fragmented using a Covaris ultrasonic disruptor (Covaris, Woburn, MA, USA). Library preparation was completed through end-repair, A-tailing, adapter ligation, purification, and PCR amplification. Initial quantification was performed with Qubit 2.0, followed by library dilution. The Agilent 2100 system was used to detect insert sizes, and once the expected size was confirmed, quantitative real-time PCR (Q-PCR) determined the library’s effective concentration for quality assurance. High-throughput sequencing was performed on an Illumina platform (HiSeq/MiSeq) after cBOT system clustering. The mean depth of sequencing was expected to reach 30×–40×.

The sequencing data were converted from Bcl to FASTQ format for raw sequence output. The data were quality-filtered and aligned to the merged reference genomes of durum wheat Svevo (*Triticum turgidum* L. ssp. *durum* Svevo, TRITD_v1) [54] and AL8/78 (*Aegilops tauschii* Coss. ssp. *strangulata*, ATGSP) [55] through BWA to generate Sam files. The ordered Bam files were produced by sorting the Sam files using Samtool Sort and removing redundancy files with Picard. The FISH results were combined and the size of the 2DS and 2AS translocation fragments in YL-429 were determined.

### 4.4. BSA Sequencing and Specific KASP Marker Design

BSA sequencing: Based on the phenotypic evaluations, 30 glaucous individuals and 30 non-glaucous individuals in the F_2_ populations of YL-724 × Chuanmai98 were selected to construct the phenotypically contrasting bulks. Ten individuals of YL-724 or Chuanmai98 were selected to construct the parental bulks. Equal-sized segments of the leaf and leaf sheath of each plant were collected at the booting stage and pooled as glaucous and non-glaucous bulks. Genomic DNA was extracted using the E.Z.N.A. Tissue DNA kit (Omega Bio-Tek, Norcross, GA, USA) and subjected to re-sequencing on the Illumina NovaSeq 6000 sequencing (PE150bp ×2 mode) platform in Shanghai BIOZERON Co., Ltd. (Shanghai, China).

SNP calling: To remove the adapter sequences and low-quality sequences, the raw paired-end reads were trimmed and quality-controlled using Trimmomatic v. 0.36 software (http://www.usadellab.org/cms/index.php?page=trimmomatic) [56]. The clean reads were aligned to the reference genome sequence of the synthetic hexaploid wheat derivatives Chuanmai104 [57] using BWA software v. 0.7.12-r1039 (http://bio-bwa.sourceforge.net/) [58]. After removing PCR-duplication reads with SAMtools v. 1.14 software (http://samtools.sourceforge.net/) [59], the unique and confident alignments were used to call SNP variants using “HaplotypeCaller” function in GATK v. 4.1.2.0 (http://www.broadinstitute.org/gatk/). The annotation of detected variations, including SNP (synonymous or non-synonymous mutations of SNPs) and InDel, was performed using ANNOVAR v. 2019Oct24 (http://www.openbioinformatics.org/annovar/) [60] by aligning with the gene annotation file of the reference genome.

High-quality SNP filtering: Based on the previously mentioned BSA sequencing results, candidate SNP markers were identified within the 0–13.8 Mb region of chromosome 2D. To ensure the quality and reliability of KASP marker development, the filtration of SNPs was carried out using the following steps: (1) genotyping missing rate: SNP loci with a genotyping missing rate exceeding 20% were excluded; (2) minor allele frequency (MAF): loci with an MAF < 0.05 were removed to avoid low-frequency alleles; (3) absence of nearby SNPs: SNP loci with no additional SNPs within 50 bp upstream and downstream were selected; (4) polymorphism information content (PIC): loci with a PIC < 0.2 were excluded to ensure sufficient polymorphism; (5) GC content: to prevent the selection of loci with extremely high or low GC content, SNP loci with a GC content between 40% and 60% within a 150 bp flanking region were selected; (6) even distribution: SNP loci were selected based on their uniform distribution across the genome to avoid clustering in specific regions. This approach enhances the overall representativeness of the selected markers, ensuring broader genomic coverage and reducing selection bias.

Specific KASP markers design and validation: After screening, evenly distributed SNPs on the translocation fragment of 2DS were selected and converted into KASP markers for genotyping. Polymorphic KASP markers were validated using the parental materials YL-724 and Chuanmai98. The KASP assay was conducted in a 384-well microplate with a final reaction volume of 2.01 µL per well, including 1 μL of sample DNA, 0.01 μL of KASP primer sets, and 1 μL of 2 × Master Mix. Genomic DNA was extracted from fresh leaf samples using the NuClean Plant Genomic DNA Kit (CWBio, Beijing, China) and diluted to 20 ng/µL with 1 × Tris-EDTA buffer. The primer sets comprised 0.002 µL of each two allele-specific forward primers and 0.006 µL of common reverse primer (synthesized by Shanghai Invitrogen Biotechnology Co., Ltd., Shanghai, China). The 2 × Master Mix for allele specific PCR V1 (HC scientific Bio (Chengdu) L.L.C, Chengdu, China) contained FAM and HEX fluorescent probes, along with standard-concentration ROX dye, making it suitable for KASP genotyping across various loci. The KASP thermal cycling program was set based on the “KASP genotyping manual” (https://www.primetech.co.jp/Portals/0/db/product/LGC/Genomics/KASP-genotyping-chemistry-User-guide.pdf) and performed on the Matrix Cycler (HC scientific Bio (Chengdu) L.L.C). The thermal cycling program used was as follows: a pre-denaturation cycle of 10 min at 95 °C; 10 cycles of 15 s at 95 °C; 1 min at 61–55 °C (drop −0.6 °C per cycle); 30–35 cycles of 15 s at 95 °C; and 1 min at 55 °C. The results of the KASP assay were scanned using the Matrix Scanner (HC scientific Bio (chengdu) L.L.C) and analyzed by using Matrix Master V.01.06.18 software (HC scientific Bio (Chengdu) L.L.C).

## Figures and Tables

**Figure 1 plants-14-01077-f001:**
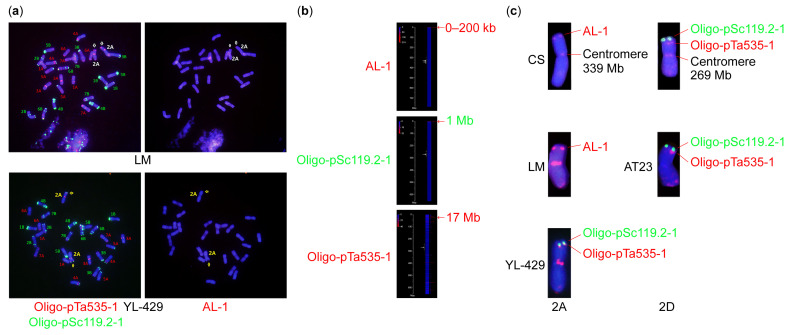
(**a**) Karyotyping using probes Oligo-pSc119.2-1 (green), Oligo-pTa535-1 (red), and AL-1 (red). (**b**) Probes’ hybridization position mapped on a physical map (reference genome: CS v1.0). (**c**) Comparison of hybridization signals.

**Figure 2 plants-14-01077-f002:**
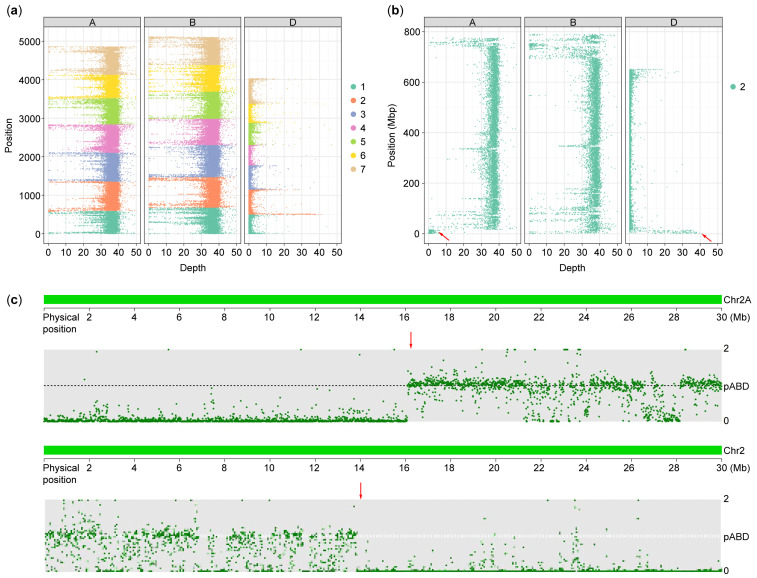
Statistical window size 100 Kbp. Red arrow shows the break point of chromosome. (**a**) Distribution of sequencing depth in each chromosome. (**b**) Distribution of sequencing depth in homoeologous chromosome group 2. (**c**) Distribution of sequencing depth in 2A and 2D chromosomes.

**Figure 3 plants-14-01077-f003:**
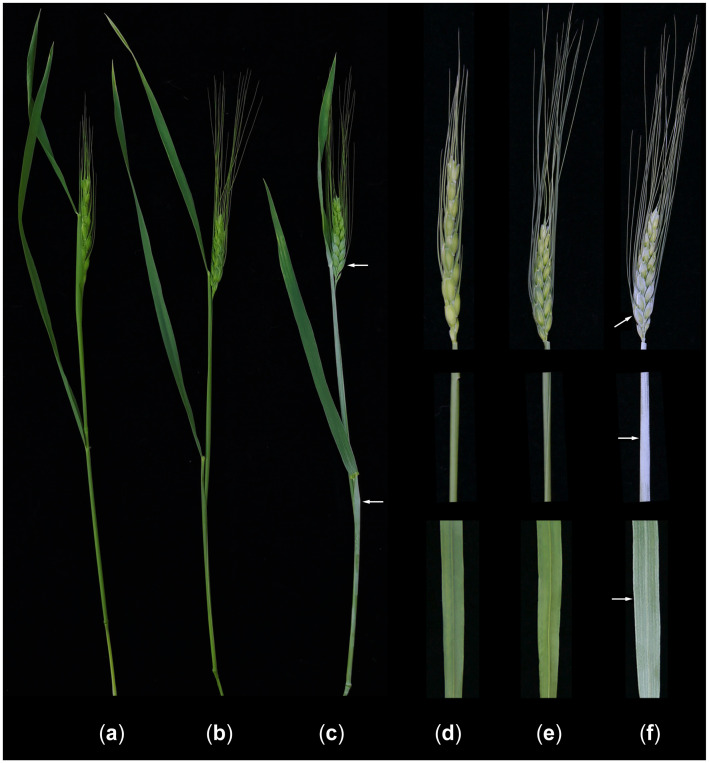
Comparison of epicuticular wax. (**a**,**d**): Synthetic hexaploid wheat YL-724 with non-glaucous epidermis. (**b**,**e**): LM T2DS-2AS.2AL translocation line YL-429 with non-glaucous epidermis. (**c**,**f**): Durum wheat LM with glaucous epidermis.

## Data Availability

The raw sequence data reported in this paper have been deposited in the Genome Sequence Archive (Genomics, Proteomics & Bioinformatics 2021) in the National Genomics Data Center (Nucleic Acids Res 2022), China National Center for Bioinformation/Beijing Institute of Genomics, Chinese Academy of Sciences (GSA: CRA022822 and CRA022975) and are publicly accessible at https://ngdc.cncb.ac.cn/gsa.

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
