# Peer review of "Identification and Specific KASP Marker Development for Durum Wheat T2DS-2AS.2AL Translocation Line YL-429 with Wax Inhibitor Gene *IW2"

_plants, 2025, doi:10.3390/plants14071077_

Round 1

Reviewer 1 Report

Comments and Suggestions for Authors

I checked your manuscript and described comments below.

Durum wheat is an important wheat used as an ingredient in bread and pasta. This paper is very significant in that it identifies the T2DS-2AS.2AL translocation line in the YL-429 lineage and develops a KASP (Kompetitive Allele Specific PCR) marker.

I don't think there are any major problems, but I think it would be better to consider the following points.

  1. I don't really understand the 27 allele specific KASP makers. I think it would be better to make a table.
  2. The URL/links for Trimmomatic v. 254 0.36 are broken. I think it would be better to include the following paper in the references.

Bolger, A. M., Lohse, M., & Usadel, B. (2014). Trimmomatic: A flexible trimmer for Illumina Sequence Data. Bioinformatics, btu170.

  1. The links for BWA, SAM tools, and ANNOVAR are not broken, but I think it would be better to include the following in the references.

BWA:

Li H. and Durbin R. (2009) Fast and accurate short read alignment with Burrows-Wheeler Transform. Bioinformatics, 25:1754-60. [PMID: 19451168]

SAM tools:

Li H.*, Handsaker B.*, Wysoker A., ​​Fennell T., Ruan J., Homer N., Marth G., Abecasis G., Durbin R. and 1000 Genome Project Data Processing Subgroup (2009) The Sequence alignment/map (SAM) format and SAMtools. Bioinformatics, 25, 2078-9. [PMID: 19505943] ANNOVAR: Wang K, Li M, Hakonarson H.

ANNOVAR:

Functional annotation of genetic variants from next-generation sequencing data Nucleic Acids Research, 38:e164, 2010

  1. For the "KASP thermal cycling program," I think it would be better to write about what you actually did rather than referring to the "KASP genotyping manual."

I don't think this paper has major problems and grammatical problems.

Author Response

Durum wheat is an important wheat used as an ingredient in bread and pasta. This paper is very significant in that it identifies the T2DS-2AS.2AL translocation line in the YL-429 lineage and develops a KASP (Kompetitive Allele Specific PCR) marker.

Comments 1:  I don't really understand the 27 allele specific KASP makers. I think it would be better to make a table.

Response 1:  Thanks for your comment. The details about the developed 27 specific KASP makers has been added in the Supplementary Table S1. Please find it in lines 155, 335, and 336.

Comments 2: The URL/links for Trimmomatic v. 0.36 are broken. I think it would be better to include the following paper in the references. (Bolger, A. M., Lohse, M., & Usadel, B. (2014). Trimmomatic: A flexible trimmer for Illumina Sequence Data. Bioinformatics, btu170.)

Response 2:  Thanks for your comment. The URL/links for Trimmomatic v. 0.36 has been corrected to ‘http://www.usadellab.org/cms/index.php?page=trimmomatic’ and the literature was cited in line 289.

Comments 3: The links for BWA, SAM tools, and ANNOVAR are not broken, but I think it would be better to include the following in the references. (BWA: Li H. and Durbin R. (2009) Fast and accurate short read alignment with Burrows-Wheeler Transform. Bioinformatics, 25:1754-60. [PMID: 19451168]); (SAM tools: Li H.*, Handsaker B.*, Wysoker A., ​​Fennell T., Ruan J., Homer N., Marth G., Abecasis G., Durbin R. and 1000 Genome Project Data Processing Subgroup (2009) The Sequence alignment/map (SAM) format and SAMtools. Bioinformatics, 25, 2078-9. [PMID: 19505943]); (ANNOVAR: Wang K, Li M, Hakonarson H. ANNOVAR: Functional annotation of genetic variants from next-generation sequencing data Nucleic Acids Research, 38: e164, 2010)

Response 3: Thanks for your advice. These literatures were cited in lines 291, 293, and 297.

Comments 4:  For the "KASP thermal cycling program," I think it would be better to write about what you actually did rather than referring to the "KASP genotyping manual."

Response 4:  Thanks for your advice. The modified KASP thermal cycling program was added in lines 329 – 331.

Reviewer 2 Report

Comments and Suggestions for Authors

The review is for a manuscript entitled "Identification and specific KASP markers development for the durum wheat T2DS-2AS.2AL translocation line with wax inhibitor gene IW2". The submitted manuscript deals with an innovative approach to improve durum wheat traits by introducing a chromosome fragment from common wheat (D genome) containing the waxing inhibitor gene Iw2. The scientific level of the research is high. The methods used - fluorescence in situ hybridization (FISH) to identify the chromosomal translocation, whole-genome sequencing to locate the translocation breakpoint, followed by bulked segregant analysis (BSA) and development of KASP markers - are state-of-the-art and relevant to the problem posed. Such a multidisciplinary set of methods testifies to a robust research approach: advanced genomics technologies have complemented classical plant cytogenetics techniques. The originality of the research lies in the characterization of new durum wheat genetic material (line YL-429 with translocation T2DS-2AS.2AL) and the identification and validation of 27 specific KASP markers coupled to the desired Iw2 allele.  This research provides new knowledge about the behavior of a foreign chromosome fragment in the durum wheat genome and practical tools to facilitate breeding selection. The obtained line without a waxy overgrowth may increase the efficiency of photosynthesis under low sunlight conditions, which may translate into higher yields in such growing regions in the future. Although the manuscript is generally well-written, I suggest a few corrections and additions that may improve its quality before publication.

To increase the usefulness of the results for other researchers and breeders, it is helpful to include a table (either in the main text or as a Supplement) containing detailed information on the 27 KASP markers developed. Such a table should consist of, at a minimum, the marker identifier, the probe sequence for both alleles (or SNP flanking sequence), and the position on the reference wheat genome. Although the text describes that these markers uniquely differentiate alleles derived from the D genome, a complete list of their sequences will be extremely valuable for potential users wishing to apply these markers in their laboratories.

I also suggest that the manuscript briefly comment on the potential genetic content of the lost 2AS segment (16.1 Mb). Based on sequencing and literature data, it is possible to estimate what genes were in the deletion on chromosome 2A. It is worth mentioning if this lost segment does not contain critical genes (or if it includes genes whose functions are known). 

Please also clarify the naming of the line to avoid possible confusion: the YL-429 line is sometimes referred to in the text by the abbreviation “LM T2DS-2AS.2AL”. It is worth making it clear the first time this name is used that it is a synonym for the YL-429 line, derived from the LM variant, which contains a translocation. This small detail will make it easier for the reader to follow the history of the material in the text.

The Latin names (italics) of the species also need to be corrected. When first used, the name should be complete (three-part: genus species, authority). 
The description of the methods lacks information about the depth of sequencing. 

The discussion should also include a few sentences on the direction of future research.

Comments on the Quality of English Language

The manuscript contains minor linguistic deficiencies.

Author Response

Comments 1:  To increase the usefulness of the results for other researchers and breeders, it is helpful to include a table (either in the main text or as a Supplement) containing detailed information on the 27 KASP markers developed. Such a table should consist of, at a minimum, the marker identifier, the probe sequence for both alleles (or SNP flanking sequence), and the position on the reference wheat genome. Although the text describes that these markers uniquely differentiate alleles derived from the D genome, a complete list of their sequences will be extremely valuable for potential users wishing to apply these markers in their laboratories.

Response 1:  Thank you for pointing this out. The details about the developed 27 specific KASP makers has been added in the Supplementary Table S1. Please find it in lines 155, 335, and 336.

Comments 2:  I also suggest that the manuscript briefly comment on the potential genetic content of the lost 2AS segment (16.1 Mb). Based on sequencing and literature data, it is possible to estimate what genes were in the deletion on chromosome 2A. It is worth mentioning if this lost segment does not contain critical genes (or if it includes genes whose functions are known).

Response 2:  Thanks for your comment. We analyzed those genes in the lost 2AS segment (16.1 Mb) and insert 2DS segment (13.8 Mb) before. For example, 894 genes were found in the lost 2AS segment including 327 high confidence genes, and 613 genes were found in the insert 2DS segment containing 350 high confidence genes. According to the GO annotation results, these high confidence genes in both regions mainly associated with terpene synthase activity and magnesium ion binding. These results will be reported in another research. We are constructing RILs population for fine mapping of the wax inhibitor gene in combination with gene expression data and the above genome resequencing results.

Comments 3:  Please also clarify the naming of the line to avoid possible confusion: the YL-429 line is sometimes referred to in the text by the abbreviation “LM T2DS-2AS.2AL”. It is worth making it clear the first time this name is used that it is a synonym for the YL-429 line, derived from the LM variant, which contains a translocation. This small detail will make it easier for the reader to follow the history of the material in the text.

Response 3:  Thanks for your comment. We have thoroughly checked the manuscript and corrected them. Please find them in lines 3, 40, 46, 127, 184, 196, 197, and 203.

Comments 4:  The Latin names (italics) of the species also need to be corrected. When first used, the name should be complete (three-part: genus, species, authority).

Response 4:  Thanks for your comment. We have thoroughly checked the manuscript and corrected them. Please find it in lines 92, 225, and 271.

Comments 5:  The description of the methods lacks information about the depth of sequencing.

Response 5:  Thank you for pointing this out. The mean depth of sequencing was expected to reach 30×- 40×. We added some descriptions in “2.2. Breakpoint identification through resequencing” (line 128), and “4.3. Genome resequencing” (lines 266 and 267).

Comments 6:  The discussion should also include a few sentences on the direction of future research.

Response 6:  Thanks for your advice. We added some description in the Discussion section. Please find it in lines 210-217.

Reviewer 3 Report

Comments and Suggestions for Authors

Comments and Suggestions for Authors

This study aims to perform chromosomal composition and translocation break-point identification of YL-429, a non-glaucous cuticular line, in Durum wheat (Triticum turgidum L. ssp. Durum) through multicolor fluorescence in situ hybridization (MC-FISH) and genomic resequencing, and to develop specific KASP markers for subsequent breeding applications. Non-glaucous wheat can reduce solar light reflection in low-light cultivation regions, enhancing photosynthetic efficiency and potentially increasing yield. For achieving the aim of the study genome resequencing was performed to identify the breakpoint of the 2D-2A chromosome translocation of YL-429; bulk sequencing analysis was conducted to detect the SNP in translocated fragment and accordingly develop specific KASP marker for breeding utilization.

The manuscript follows the requirements of “Plants” for its construction. The applied research methods are appropriate for achieving the aim of the study. The results are well illustrated and described.

A following remark can be made:

According to the template form of “Plants”, the manuscript should to end with “Conclusions” section /after “Materials and Methods”/. There is no such part in the manuscript.

In conclusion, this manuscript is recommended for publication in "Plants" after consideration of the comments made.

Author Response

Comments 1:  The manuscript follows the requirements of “Plants” for its construction. The applied research methods are appropriate for achieving the aim of the study. The results are well illustrated and described.

Response 1:  Thank you for taking time to review our manuscript.

Comments 2:  According to the template form of “Plants”, the manuscript should to end with “Conclusions” section /after “Materials and Methods”/. There is no such part in the manuscript.

Response 2:  Thanks for your comment. We added short description in the Discussion section. Please find it in lines 210-217.

Comments 3:  In conclusion, this manuscript is recommended for publication in "Plants" after consideration of the comments made.

Response 3:  Thank you for taking time to review our manuscript.